# Stability of Strained Stanene Compared to That of Graphene

**DOI:** 10.3390/ma15175900

**Published:** 2022-08-26

**Authors:** Igor V. Kosarev, Sergey V. Dmitriev, Alexander S. Semenov, Elena A. Korznikova

**Affiliations:** 1Research Laboratory for Metals and Alloys under Extreme Impacts, Ufa State Aviation Technical University, 450008 Ufa, Russia; 2Mechanical Engineering Research Institute of the Russian Academy of Sciences–Branch of Federal Research Center “Institute of Applied Physics of RAS”, 603024 Nizhny Novgorod, Russia; 3Center for Design of Functional Materials, Bashkir State University, 450076 Ufa, Russia; 4Polytechnic Institute Mirny Branch, North-Eastern Federal University, 678170 Mirny, Russia; 5Institute of Oil and Gas Engineering and Digital Technologies, Ufa State Petroleum Technological University, 450064 Ufa, Russia

**Keywords:** stanene, graphene, elastic strain, stability, phase transition, molecular dynamics

## Abstract

Stanene, composed of tin atoms, is a member of 2D-Xenes, two-dimensional single element materials. The properties of the stanene can be changed and improved by applying deformation, and it is important to know the range of in-plane deformation that the stanene can withstand. Using the Tersoff interatomic potential for calculation of phonon frequencies, the range of stability of planar stanene under uniform in-plane deformation is analyzed and compared with the known data for graphene. Unlike atomically flat graphene, stanene has a certain thickness (buckling height). It is shown that as the tensile strain increases, the thickness of the buckled stanene decreases, and when a certain tensile strain is reached, the stanene becomes absolutely flat, like graphene. Postcritical behaviour of stanene depends on the type of applied strain: critical tensile strain leads to breaking of interatomic bonds and critical in-plane compressive strain leads to rippling of stanene. It is demonstrated that application of shear strain reduces the range of stability of stanene. The existence of two energetically equivalent states of stanene is shown, and consequently, the possibility of the formation of domains separated by domain walls in the stanene is predicted.

## 1. Introduction

In addition to graphene [1], other single-element two-dimensional materials with a similar hexagonal structure have been obtained and studied. These are group IV elements, such as silicene, germanene, and stanene (2D-Xenes). In the past few years, they have attracted the attention of researchers due to a combination of electronic and optical properties that can be used in nanotechnology [2,3,4,5].

Unlike atomically flat graphene, all two-dimensional group IV materials (silicene, germanene, and stanene) are buckled and have a certain thickness (buckling height [6,7]). The free-standing stanene has a thickness of h=0.85÷0.90 Å according to density functional theory (DFT) calculations [8,9]. Stanene grown on a substrate may be thinner or thicker than a free-standing stanene.

The first successful attempt to synthesize stanene was made on the Bi_2_Te_3_(111) surface in 2015 [10] and later on many other substrates using the epitaxial methods of growth [4,11]. The thickness of stanene on Bi_2_Te_3_(111) is reported to be h=1.2 Å [10], which is larger than that of free-standing stanene. From the DFT calculations, the structure of the stanene with dumbbell units with a thickness of 3.41 Å was predicted [12]. Large area stanene can be epitaxially grown on Ag(111) with h=0.12 Å [13,14]. A nearly flat stanene-like honeycomb structure with three Sn atoms per unit cell was epitaxially grown on an Au(111) surface [15], and ultraflat stanene (h=0) was epitaxially grown on a Cu(111) surface [16]. Strain-free stanene with h=0.20 Å was grown on a Pd(111) surface terminated by a Pd_2_Sn(111) surface alloy [17]. In this study, it will be shown that when stretched, a free-standing stanene, modelled by the Tersoff potential, can transform into a flat configuration (h=0).

Stanene-based gas sensors are described in the review [18]. The adsorption energies of small molecules on stanene were calculated using the ab initio methods [19]. The biaxial strain of the stanene increases the adsorption energy of NO_2_ molecules, thereby improving the sensitivity to the hazardous gas [20]. The absorption of toxic molecules SO_2_ and H_2_S on the stanene can be improved with the help of defects and dopants [21]. The B-doped stanene better absorbs O_3_ and SO_3_ molecules [22]. The stable configurations, adsorption energies, electronic properties, and charge transfer of toxic NO_2_, SO_2_, and NH_3_ and non-toxic CO_2_ molecules on stanane are reported in the work [23].

Elastic strain engineering is a powerful technique for modifying the properties of nanomaterials because, compared to conventional materials, they can withstand much higher stresses and deform elastically up to large strains [24,25,26]. At large elastic deformations, the lattice symmetry changes, and accordingly, the electronic structure changes, which leads to new physical, chemical, and mechanical properties of the material [27]. Changes in the bond length and bond angle as the functions of tensile strain along the zigzag or armchair direction, as well as the G-mode splitting with strain, were discussed for graphene in [28].

As mentioned above, stretched stanene adsorbs NO_2_ molecules better [20]. Biaxial strain can effectively tune the bandgap of stanene nanomeshes [27]. Ferroelectric and ferroelastic properties of monolayer group IV monochalcogenides can be effectively modulated by elastic strain engineering [29]. The electronic properties of nanomembranes of the topological crystalline insulators SnTe and Pb_1−*x*_Sn_*x*_(Se,Te) are easily tuned using elastic strain engineering, which makes it possible to tune band gap and obtain giant piezoconductivity [30]. Elastic strain engineering can be used to tailor single photon emission properties of wrinkled WSe_2_ monolayers [31]. Biaxial tension applied to silicene, germanene and stanene hardens long wavelength out-of-plane acoustic phonons, which leads to an increase in thermal conductivity [32]. A similar effect was also described for twisted carbon nanoribbons [33].

It is now generally accepted that numerical methods are very useful in the study of structure and properties of nanomaterials [34,35,36].

Flat as well as buckled stanene under tension along the zigzag and armchair directions was modelled by calculating the second, third, fourth, and fifth order elastic moduli tensors by using first principles calculations [6]. It was shown that the buckling height of buckled stanene decreases with tension but does not vanish within the stability range of the material. The ideal strength in the armchair and zigzag directions for flat and buckled stanene was also analysed, and it was found that the calculated critical strain is about 0.23 for both directions.

The effect of strain on the electronic and mechanical properties of stanene was considered in the work [37]. The density functional theory and molecular mechanic models were used to find the in-plane stiffness of stanene to be 40 N/m. Tension of 2% reduced the buckling height by about 4% from the initial value of h=0.86 Å. Ideal strength is defined as the maximum stress that a defect-free material can withstand [38]. In a theoretical work [39], using DFT simulations, it was shown that the ideal strength of 2D stanene reaches E/7.4, where *E* is Young’s modulus in tension. This value exceeds Griffith’s estimate of the theoretical strength E/9. This inconsistency can be explained by the too small thickness used by the authors to estimate Young’s modulus [7]. The effect of temperature and strain rate on mechanical properties of stanene under uniaxial and biaxial strain was addressed in [40] using the molecular dynamics method. The failure of the stanene was observed at a strain of about 20% and a stress of 3 GPa under biaxial tension and tension along the zigzag direction at a strain rate of 10^7^ s^−1^ with slightly lower critical values for tension along the armchair direction.

Elastic properties and crack propagation in single and bilayer stanene were analyzed based on molecular dynamics simulations using Tersoff bond order potential [41]. At the strain rate of 10^8^ s^−1^, failure of stanene was observed at about 38% strain, and 15 GPs stress for tension along the armchair direction and lower critical values were found for tension along the zigzag direction (33% strain and 14 GPa stress). A comparative analysis of the mechanical properties of 2D-Xenes suggests that stanene is the weakest and softest member of this family [42].

Formulas for the shear modulus of hexagonal nanostructures were derived based on a physically motivated model [43]. In the works [44,45,46], concrete examples of materials with unusual properties are considered, and the origin of these properties at the discrete level is described. Electronic properties of bilayer stanene can be improved via modulating the stacking order and angle of the bilayers [47].

In order to take full advantage of the possibilities of elastic strain engineering, it is necessary to know the stability region of the stanene in the space of the strain tensor components $\varepsilon_{xx}$, $\varepsilon_{yy}$, and $\varepsilon_{xy}$. For graphene, this problem was solved in [48] using the potentials developed by Savin [49]. This problem is difficult to solve experimentally because it is not easy to apply high enough stress to break a defect-free stanene. In such circumstances, theoretical studies make it possible to shed light on many properties of two-dimensional structures, the study of which by experimental methods is difficult. However, theoretical studies also encounter certain difficulties. For example, ab initio methods, due to the high demands on computer resources, consider a small number of atoms in short periods of time [6,50,51,52], and the molecular dynamics method gives results that depend on the interatomic potentials [53,54,55,56]. Despite this, modeling allows at least a qualitative characterization of many important properties of nanomaterials.

In this work, we aim to calculate the range of stability of plane stanene modelled with the Tersoff potential in the space of the strain tensor components $\varepsilon_{xx}$, $\varepsilon_{yy}$, and $\varepsilon_{xy}$.

## 2. Materials and Methods

Molecular dynamics is used to simulate the stability of stanene.

### 2.1. Structure of Stanene

A free-standing, infinitely large, defect-free, single-sheet stanene is considered. According to theoretical and experimental data [57,58,59,60,61], undeformed, free-standing stanene has a hexagonal buckled structure. A primitive translational cell of stanene defined by the translation vectors $a_{1}$ and $a_{2}$, is shown in Figure 1a projected onto the XY plane. The *X* and *Y* axes of the Cartesian coordinate system are oriented in the zigzag and armchair directions, respectively. Atoms 1 and 2 represent two triangular sublattices colored black and red; generally speaking, they have different *Z*-coordinates. Figure 1b shows three valence bonds, 1, 2 and 3, and six valence angles, $\varphi_{i}$, and $\tau_{i}$, i=1,2,3. These structural parameters will be analyzed for homogeneously deformed stanene.

Figure 1c shows the side views of buckled stanene having thickness *h* projected onto the XZ and YZ planes. In Figure 1d, similar side views are shown for the absolutely flat stanene.

### 2.2. Structure Generation and Homogeneous Deformation Application

The structure of stanene is generated by translation of a pair of atoms by the vectors $a_{1}=a\left(1,0,0\right)$ and $a_{2}=\left(a/2\right)\left(1,\sqrt{3},0\right)$, where *a* is the lattice parameter (see Figure 1a). The length of the translation vectors $|a_{1}\left|=\right|a_{2}|=a$ can be obtained by projecting the equilibrium bond length $\rho_{0}$ onto the XY plane. The radius–vectors of Sn atoms in unstrained structure are $R_{m,n,k}^{0}=ma_{1}+na_{2}+s_{k}$, where the integers m,n define the primitive cell number, and k=1,2 is the number of atoms in a primitive translational cell. The vectors $s_{k}$, k=1,2, in unstrained stanene are taken as $s_{1}=\left(0,0,0\right)$, $s_{2}=\left(a/2,a/2\sqrt{3},h\right)$, where *h* is the structure thickness depicted in Figure 1c.

Stanene is considered under homogeneous plain stress conditions achieved by application of in-plane strain with the components $\varepsilon_{xx},\varepsilon_{yy},\varepsilon_{xy}$. The equilibrium positions of atoms in homogeneously strained stanene are $R_{m,n,k}^{s}=mp_{1}+np_{2}+q_{k}$, where vectors $p_{1}=a_{1}+a_{1}T,\;p_{2}=a_{2}+a_{2}T$ are defined through the matrix
(1)$$\begin{array}{c} T=\left(\begin{array}{cc} \varepsilon_{xx} & \varepsilon_{xy}/2\\ \varepsilon_{xy}/2 & \varepsilon_{yy} \end{array}\right). \end{array}$$

Vectors $q_{k}$ describe the position of atoms in a primitive cell after deformation. By the rigid shift of the structure in space, one can always take $q_{1}=\left(0,0,0\right)$. On the other hand, the equilibrium position of atom 2 in the primitive cell, $q_{2}$, must be obtained by relaxation of the structure. As a result of relaxation, the equilibrium thickness of the deformed structure will also be determined.

The relaxation is performed by the steepest descent method using self-made code written in the C++ algorithmic language. Relaxation stops when the value of the maximum force acting on the atoms becomes less than 10^−8^ eV/Å.

### 2.3. Tersoff Potential

The many-body Tersoff potential [62] is used to describe interactions between tin atoms in stanene. In Tersoff’s models, the potential energy of the system is taken as
(2)$$\begin{array}{c} U_{s}=\frac{1}{2}\sum_{i}\sum_{j\neq i}f_{c}\left(r_{ij}\right)\left[a_{ij}f_{r}\left(r_{ij}\right)+b_{ij}f_{a}\left(r_{ij}\right)\right], \end{array}$$
where $r_{ij}=\left|R_{i}-R_{j}\right|$ is the distance between atoms *i*, and *j* ($R_{i}$ is the radius-vector of atom *i*). The function $f_{c}\left(r_{ij}\right)$ limits the range of interaction to the nearest neighbors. This function is taken in the form
(3)$$\begin{array}{c} f_{c}\left(r\right)=\left\{\begin{array}{cc} 1, & \;r<R-D;\\ \frac{1}{2}-\frac{1}{2}\sin\frac{\pi (r-R)}{2D}, & \;R-D\leq r<R+D;\\ 0, & \;r\geq R+D. \end{array}\right. \end{array}$$
where *R* and *D* are the constants describing the cut-off radius of the potential.

The potential energy (2) includes the embeddings of repulsive and attractive forces,
(4)$$\begin{array}{c} f_{r}\left(r_{ij}\right)=Ae^{-\lambda_{1}r_{ij}},\;f_{a}\left(r_{ij}\right)=-Be^{-\lambda_{2}r_{ij}}, \end{array}$$
where *A*, *B*, $\lambda_{1}$, and $\lambda_{2}$ are the potential parameters describing the dimer strength and Pauling constants [63,64].

The component $a_{ij}$ for the repulsive force is taken equal to 1 [65].

The $b_{ij}$ term in (2) describes the weakening of the bond between atoms *i* and *j* due to the presence of another bond between atoms *i* and *k*. This term describes the energy of valence angles,
(5)$$\begin{array}{c} b_{ij}={\left(1+\beta^{N}\zeta_{ij}^{N}\right)}^{\frac{-1}{2N}}, \end{array}$$
(6)$$\begin{array}{c} \zeta_{ij}^{n}=\sum_{k\neq i,j}f_{c}\left(r_{ik}\right)g\left(\theta_{ijk}\right)e^{\left[\lambda_{3}^{M}{\left(r_{ij}-r_{ik}\right)}^{M}\right]}, \end{array}$$
(7)$$\begin{array}{c} g\left(\theta_{ijk}\right)=\gamma \left(1+\frac{c^{2}}{d^{2}}-\frac{c^{2}}{d^{2}+{\left(\cos\theta_{ijk}-\delta \right)}^{2}}\right). \end{array}$$

Parameters β, *N*, $\lambda_{3}$, *c*, *d*, and δ are adjustable, while M=3.0 and γ=1.0 are constants. In this work, a more symmetrical form of the potential is used in which $\bar{b_{ij}}=\frac{b_{ij}+b_{ji}}{2}$.

Parameters of the Tersoff potential are listed in Table 1.

The Hamiltonian of the system (total energy of the computational cell is)
(8)$$H=U_{s}+\frac{M}{2}\sum_{i}{\left(\frac{dR_{i}}{dt}\right)}^{2},$$
where $U_{s}$ is the potential energy of the system defined by Equation (2), and the second term on the right-hand side of the Hamiltonian gives the kinetic energy (*M* is the mass of the stanene atom). The equations of motion are derived from the Hamiltonian Equation (8) using the Hamilton’s principle.

Let the computational cell include $N_{x}\times N_{y}$ primitive translational cells, each of which contains two atoms. Atoms are conveniently numbered with three indices $m=1,...,N_{x}$, $n=1,...,N_{y}$, and k=1,2, as described in Section 2.2.

### 2.4. Check for Stability of Strained Stanene

For strained stanene, we check its stability by calculating the frequency of phonon modes. To do so we write down the equations of motion for the two atoms of the (m,n)th translational cell
(9)$$\begin{array}{c} M\frac{d^{2}u_{m,n,k}}{dt^{2}}=-\frac{\partial P_{m,n}}{\partial u_{m,n,k}},\;\;k=1,2. \end{array}$$

Then we linearize these equations and look for their solutions in the form
(10)$$\begin{array}{c} u_{m,n,k}\left(t\right)=e\;\exp\left[i\left(q_{x}m+q_{y}n-\omega t\right)\right],\;\;k=1,2. \end{array}$$

In Equations (9) and (10), $u_{m,n,k}$ is the displacement vector of (m,n,k)th atom from equilibrium position, and it is assumed that $|u_{m,n,k}|\ll \rho_{0}$; *M* is the mass of tin atom; $P_{m,n}$ is the potential energy of (m,n)th primitive cell. Six components of the displacement vectors $u_{m,n,1}$ and $u_{m,n,2}$ define positions of two atoms in a primitive translational cell; e is the normalized six component eigenvector (e,e)=1; *i* is an imaginary unit; $0\leq q_{x},q_{y}<2\pi$ are the components of the wavevector; ω is the phonon frequency.

Substitution of Equation (10) into linearized equations of motion results in the eigenproblem whose solution gives eigenvalues $\omega_{j}^{2}$ and corresponding eigenvectors $e_{j}$, j=1,...,6. A flat stanene is stable if all eigenvalues $\omega_{j}^{2}$ in the entire first Brillouin zone in the phonon spectrum are positive. The first Brillouin zone is scanned with $∆q_{x}$ and $∆q_{y}$ steps equal to π/100. The eigenvalue problem mentioned above is solved at each point, and if all $\omega_{j}^{2}$, j=1,...,6 are all positive, then the flat structure is stable. If the spectrum has negative eigenvalues, then the planar state is unstable.

The calculation of dispersion curves and stability analysis for stanene are performed using a home-made C++ code that implements the solution of the eigenvalue problem based on the iterative Jacobi algorithm for calculating the eigenvalues and eigenvectors of a real symmetric matrix. The iterations stop when the maximum off-diagonal element of the stiffness matrix becomes less than 10^−5^ eV/Å^2^.

## 3. Results and Discussion

Let us present the numerical results obtained for the equilibrium structure of stanene and its stability.

### 3.1. Equilibrium State of Stanene

Before applying deformation, it is necessary to find the equilibrium state of the structure relative to which deformation will be applied. To do this, we change the lattice parameter *a* and independently the height of the structure *h* in order to find the state of minimum energy. The process of finding this state is illustrated in Figure 2.

The equilibrium state of stanene is characterized by the equilibrium bond length $\rho_{0}\approx 2.88$ Å, structure height h=0.99 Å, potential energy per atom U≈−3.31 eV, and the lattice parameter a=2.70 Å. Recall that according to the DFT calculations [8,9], the free-standing stanene has somewhat smaller thickness of h=0.85÷0.90 Å.

Let us estimate the Poisson’s ratio of the undeformed mill. To do this, we calculate the energy of elastic deformation per atom for three deformation states:

Let us estimate the Poisson’s ratio of the undeformed stanene. To do this, we calculate the elastic strain energy per atom for the three strain states, state 1: $\varepsilon_{xx}=\varepsilon$, $\varepsilon_{yy}=0$; state 2: $\varepsilon_{xx}=0$, $\varepsilon_{yy}=\varepsilon$; and state 3: $\varepsilon_{xx}=\varepsilon_{yy}=\varepsilon$, with $\varepsilon_{xy}=0$ in all cases. The elastic energy density for the plain stress conditions can be expressed through the elastic constants and strain components as follows
(11)$$U=\frac{E}{2(1-\nu^{2})}\left(\varepsilon_{xx}^{2}+\varepsilon_{yy}^{2}+2\nu \varepsilon_{xx}\varepsilon_{yy}+\frac{1-\nu }{2}\varepsilon_{xy}^{2}\right),$$
where *E* and ν are the Young modulus and Poisson’s ratio. Substituting the values of strain for the three considered strain states and eliminating *E*, one finds
(12)$$\nu =\frac{U_{3}}{2U_{1}}-1=\frac{U_{3}}{2U_{2}}-1.$$

The result for $\varepsilon =10^{-3}$ is $U_{1}=3.87\times 10^{-6}$, $U_{2}=3.86\times 10^{-6}$, and $U_{3}=9.61\times 10^{-6}$ eV. The almost equal values of $U_{1}$ and $U_{2}$ confirm that the stanene is elastically isotropic at small strains. Substituting the values of energy into Equation (12), one finds ν≈0.242≈0.245. DFT results give larger values of Poisson’s ratio. In the work [67], $\nu_{\mathrm{zigzag}}=0.42$, $\nu_{\mathrm{armchair}}=0.36$, and in [68], ν=0.395.

It should be noted that in molecular dynamics calculations, Poisson’s ratio is often found with a larger uncertainty than Young’s modulus. For example, in the review [69], dozens of papers on the elastic properties of graphene are analyzed, and the following ranges of calculated Young’s modulus E=312÷384 N/m and Poisson’s ratio ν=0.16÷0.46 are indicated.

### 3.2. Phonon Dispersion Curves

Our stability analysis is based on the calculation of the frequencies of small-amplitude oscillations of stanene. It is useful to calculate the phonon dispersion curves of undeformed stanene and compare them with the available results. Our results for the Tersoff potential are shown in Figure 3 and can be compared with the DFT-based results reported in [32], see Figure 1(c) of their work. There are three acoustic and three optic branches of the dispersion curves. We note a reasonably good agreement of the results. The maximum phonon frequency is 6.2 THz for the Tersoff potential and about 6.0 THz from ab initio calculations [32]. The ZA and ZO modes with out-of-plane displacements of atoms have the lowest acoustic and optical frequencies, respectively. Longitudinal phonons (LA and LO) have frequencies higher than transverse phonons (TA and TO, respectively) because the shear modulus is less than the tensile modulus.

### 3.3. Stability Region of Planar Stanene under In-Plane Tensile and Compressive Strain

Figure 4 shows the stability region of a planar stanene subjected to biaxial normal strain $\varepsilon_{xx}$ and $\varepsilon_{yy}$ at $\varepsilon_{xy}=0$. A flat stanene is stable inside the region shown by the solid line ABCDEFA and unstable outside this region. The postcritical behavior depends on which boundary of the stability region is crossed when leaving the stability region. When the CD line is crossed, atomic bonds of type 1 and 3 are broken, shown in Figure 1a. If the DE line is crossed, then the type 2 bonds break as they are most loaded in this case. If line ABC is crossed, then the stanene sheet experiences planar compression along the *Y* axis; hence ripples are formed that are elongated along the *X* axis as the planar configuration becomes unstable. Similarly, when crossing the AFE line, ripples oriented along the *Y* axis appear in the stanene compressed along the *X* axis.

Interestingly, within the region of stability, stanene exists in two different states. For relatively small tensile strain, stanene is buckled (h>0), and it becomes absolutely flat (h=0) under sufficiently large tensile strain. These two regions are separated by the dashed line $\varepsilon_{yy}=-1.03\varepsilon_{xx}+0.272$ in Figure 4.

The change in the height of stanene in the range of stability of the planar stanene is shown in Figure 5a. It can be seen that *h* decreases with increasing tensile strain $\varepsilon_{xx}$ and $\varepsilon_{yy}$ and is equal to zero in the BCDEFB region.

Possible values of stanene height along the line $\varepsilon_{xx}=\varepsilon_{yy}$ are shown in Figure 5b. The negative value of *h* means that the red and black sublattices shown in Figure 1 change the sign of the *Z* coordinate, Z→−Z. Suppose we have obtained an absolutely flat stanene with a sufficiently large tensile strain. Then a decrease in the tensile strain will lead to an approach to the bifurcation point in which the system has two energetically equivalent paths of development. In the region of nonzero *h*, atoms belonging to the red and black sublattices can have coordinates Z>0 and Z<0, respectively, or vice versa. In the first case, h>0, and in the second case, h<0. The red dashed line in Figure 5b shows an absolutely flat stanene, which is unstable after crossing the bifurcation point as the tensile strain decreases.

The fact of the existence of two energetically equivalent states implies the possibility of the existence in stanene of domains separated by domain walls.

### 3.4. Comparison of the Stability Regions of Stanene and Graphene

In the literature, the stability regions of graphene were described [48,70] using the Savin potential [49] and the modified Brenner potential [65,71], and here we compare our results for stanene with existing results for graphene. Note that the accuracy of various interatomic potentials for graphene have been assessed in the work [54]. It was proved that the Savin potential reproduces many properties of carbon structures well [72,73,74,75].

A comparison of the stability regions of stanene and graphene is shown in Figure 6. At first glance, the stability regions of stanene and graphene are similar, but there are some important differences between them. The most striking difference is that stanene exists in a buckled and absolutely flat configuration, while graphene is flat throughout its stability range.

Another important difference is that buckled stanene allows very little negative strain, unlike graphene, which remains flat in a range of negative values of $\varepsilon_{xx}$ at large positive values of $\varepsilon_{yy}$ and in a range of negative values of $\varepsilon_{yy}$ for large positive $\varepsilon_{xx}$. Note that absolutely flat stanene, like graphene, has regions of stability under negative strains. Based on this, it can be concluded that absolutely flat stanene has similar properties to graphene, while buckled stanene behaves differently. This can be explained by the difference in Poisson’s ratio of buckled stanene and graphene.

The fact that the stanene is stable under greater biaxial tension with $\varepsilon_{xx}=\varepsilon_{yy}$ can be explained by stanene stretching due to flattening.

### 3.5. Region of Stability of Planar Stanene and Graphene in the Presence of Shear Strain

The effect of shear strain on the stability regions of graphene and stanene is presented in Figure 7a and Figure 7b, respectively. The borders of stability regions of planar structures are shown as black, red, and blue lines for $\varepsilon_{xy}$ = 0.0, 0.15 and 0.3, respectively.

Both graphene and stanene stretched along the armchair direction (i.e., along the *Y* axis) retain greater stability when shear strain is applied. However, as $\varepsilon_{xy}$ increases, the stability region of graphene narrows along the strain component $\varepsilon_{xx}$ and shrinks towards the center, while the stability region of stanene reduces from the right and from the bottom.

It is also interesting to look at the structure of strained stanene; for this, two points, A and B, shown in Figure 7b were chosen. Point A corresponds to biaxial strain $\varepsilon_{xx}=\varepsilon_{yy}=0.118$ and point B to uniaxial strain $\varepsilon_{xx}=0,\varepsilon_{yy}=0.3$. The effect of addition of shear strain $\varepsilon_{xy}$ on the stanene structure will be analyzed.

In Figure 8a, it can be observed that the structure at $\varepsilon_{xy}=0$ retains the hexagonal symmetry under biaxial tension. This symmetry is lost after application of shear stain, see (c) and (e). Figure 8 also shows that the height of the structure decreases with increasing shear strain, cf. (b) with (d) and (f).

In Figure 9a, it can be seen that under tension along the armchair direction with zero shear strain the hexagons are elongated along the *Y* axis. In (c) and (e), in the presence of the shear strain component, the elongated hexagons are tilted. In point B, stanene is absolutely flat for any value of the shear strain, see (b), (d), and (f).

Parameters of the structures at points A and B marked in Figure 7b are presented in Table 2 for different values of shear strain. Valence bonds and valence angles are defined in Figure 1b. It can be seen that at point A for $\varepsilon_{xy}=0$ all valence angles are equal and all valence bonds have the same length. The equality disappears when a shear strain is applied. At point B at $\varepsilon_{xy}=0$ all three bond angles are different, and all three bonds have different lengths.

The penultimate column of Table 2 shows that the potential energy per one atom is greater for stanene at point B compared to point A. The last column of Table 2 shows the thickness of the stanene for all considered strain states. In point B, stanene is absolutely flat for all three considered strain states.

## 4. Conclusions and Future Work

By finding the vibration frequencies of tin atoms in equilibrium stanene, the stability region of a planar structure was found in the space of strain components $\varepsilon_{xx}$, $\varepsilon_{yy}$, and $\varepsilon_{xy}$, see Figure 4 and Figure 7b. The results were compared to the stability region of homogeneously strained graphene reported in [48,70], see Figure 6 and Figure 7a. Stanene was simulated with the use of the Tersoff potential [62] and graphene with the Savin [49] and Brenner [65,71] potentials. The accuracy of the potentials was discussed in [54,72,73,74,75]. The accuracy of the potential at not very large atomic displacements can be estimated by comparing the phonon dispersion curves shown in Figure 3 with those calculated on the basis of the DFT theory [32]. As already mentioned, the agreement between the dispersion curves calculated using the Tersoff potential and those calculated ab initio is quite good. The accuracy of the Tersoff potential at large values of strain may not be as good because the dispersion curves are calculated for the linearized equations of motion.

It was demonstrated that the stanene is buckled, has a nonzero thickness *h* under a small stretch, and becomes absolutely flat under a sufficiently large stretch, namely, for $\varepsilon_{yy}>-1.03\varepsilon_{xx}+0.272$, see Figure 4. Stanene height as the function of $\varepsilon_{xx}$ and $\varepsilon_{yy}$ is presented in Figure 5. Actually, Figure 5b shows the bifurcation diagram where negative *h* means that the red and black sublattices shown in Figure 1 change the sign of the *Z* coordinate, Z→−Z. The fact of the existence of two energetically equivalent states (with positive and negative *h*) implies the possibility of the existence in stanene of domains separated by domain walls.

In a future work, the combined effect of deformation and temperature on the stability of stanene will be analyzed (in this work, the temperature effect was neglected). The structure and energy of domain walls, the existence of which was predicted in this work, will be investigated. Postcritical ripples in stanene under in-plane compression will be studied in accordance with the works [36,70,76]. Of interest are also the mechanical properties of stanene nanotubes and nanotube bundles similar to their carbon counterparts [77,78,79]. The propagation of a shock wave in stanene may have interesting features due to the nonzero thickness [80].

## Figures and Tables

**Figure 1 materials-15-05900-f001:**
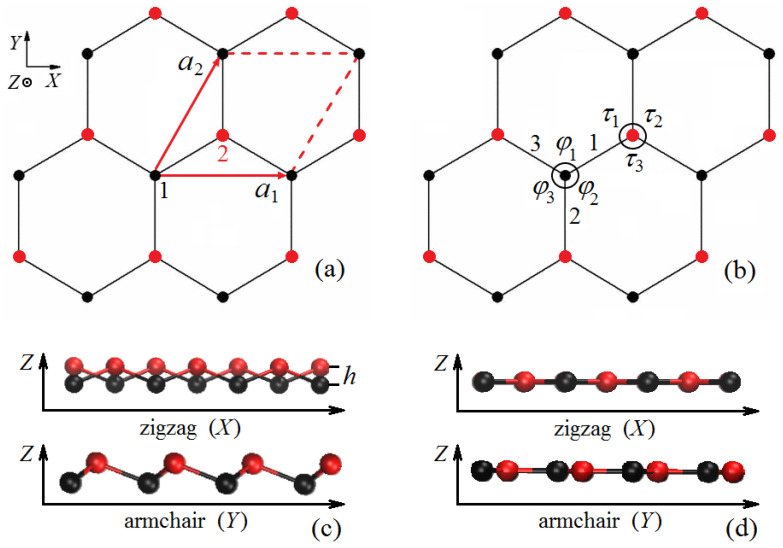
(**a**) The structure of the stanene shown in projection onto the XY plane with the *X* and *Y* axes oriented in the zigzag and armchair directions, respectively. A primitive translational cell containing two Sn atoms is defined by the translation vectors $a_{1}$ and $a_{2}$. The atoms of the two sublattices are colored black and red and can have different *Z*-coordinates. (**b**) Valence bonds and angles that will be analyzed when the stanene is deformed. Panels (**c**) and (**d**) show the stanene projected onto the XZ and YZ planes for the buckled structure having thickness *h* and absolutely flat structure, respectively.

**Figure 2 materials-15-05900-f002:**
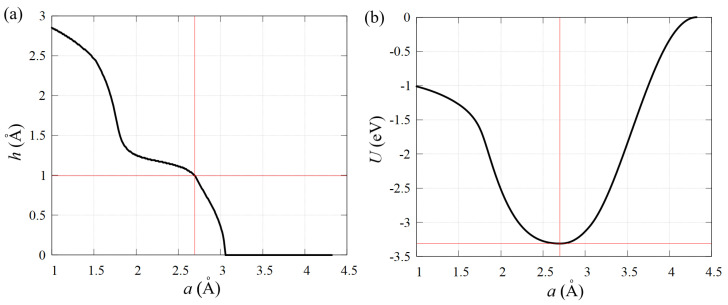
Dependence of (**a**) stanene height *h* and (**b**) potential energy per atom *U* on the lattice parameter *a* during relaxation with preservation of the hexagonal structure to find the equilibrium state. The red lines indicate the equilibrium state of the stanene structure.

**Figure 3 materials-15-05900-f003:**
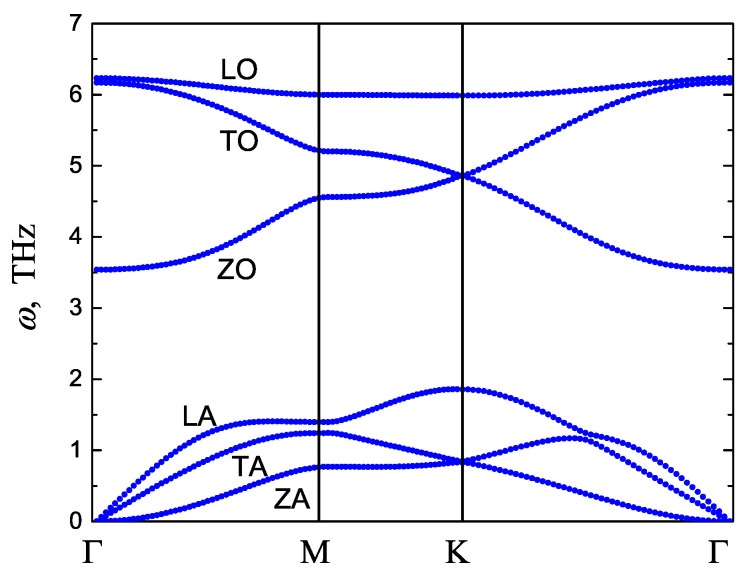
Phonon dispersion curves of unstrained stanene.

**Figure 4 materials-15-05900-f004:**
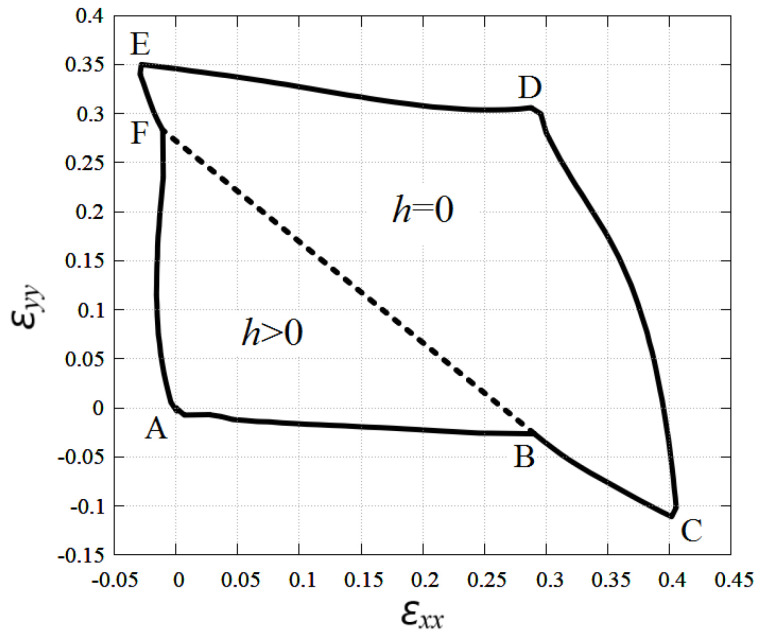
Stability region of planar stanene under normal strain components $\varepsilon_{xx}$ and $\varepsilon_{yy}$ at $\varepsilon_{xy}=0$. Inside the region ABCDEFA, planar stanene is stable. Dotted line separates the regions of buckled stanene (h>0) and absolutely flat stanene (h=0). The transition line is $\varepsilon_{yy}=-1.03\varepsilon_{xx}+0.272$.

**Figure 5 materials-15-05900-f005:**
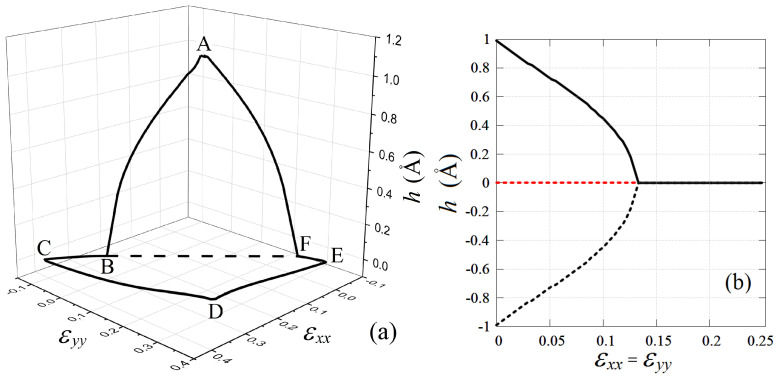
(**a**) Stanene height change in the planar stanene stability region. In the ABFA region, stanene is buckled (h>0), and in the CDEC region, it is absolutely flat (h=0); (**b**) bifurcation diagram showing a possible change in the height of the stanen along the line $\varepsilon_{xx}=\varepsilon_{yy}$.

**Figure 6 materials-15-05900-f006:**
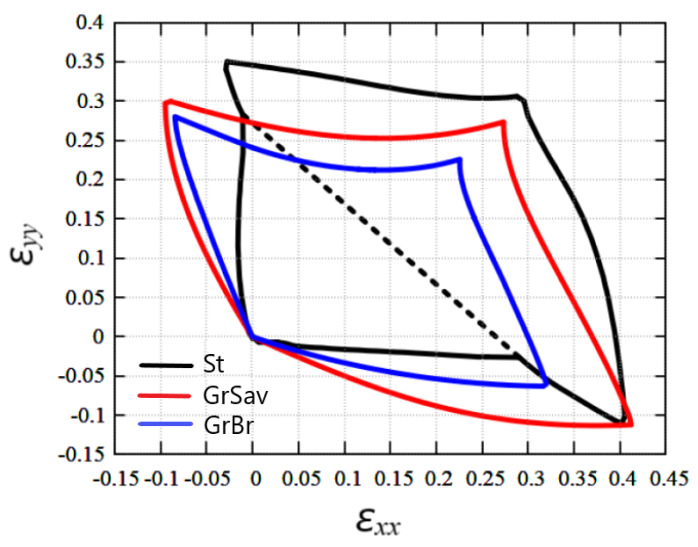
Comparison of stability regions of planar stanene (St, black line) and graphene obtained using Savin’s potential [49] (GrSav, red line) and Brenner’s potential (GrBr, blue line) [65,71].

**Figure 7 materials-15-05900-f007:**
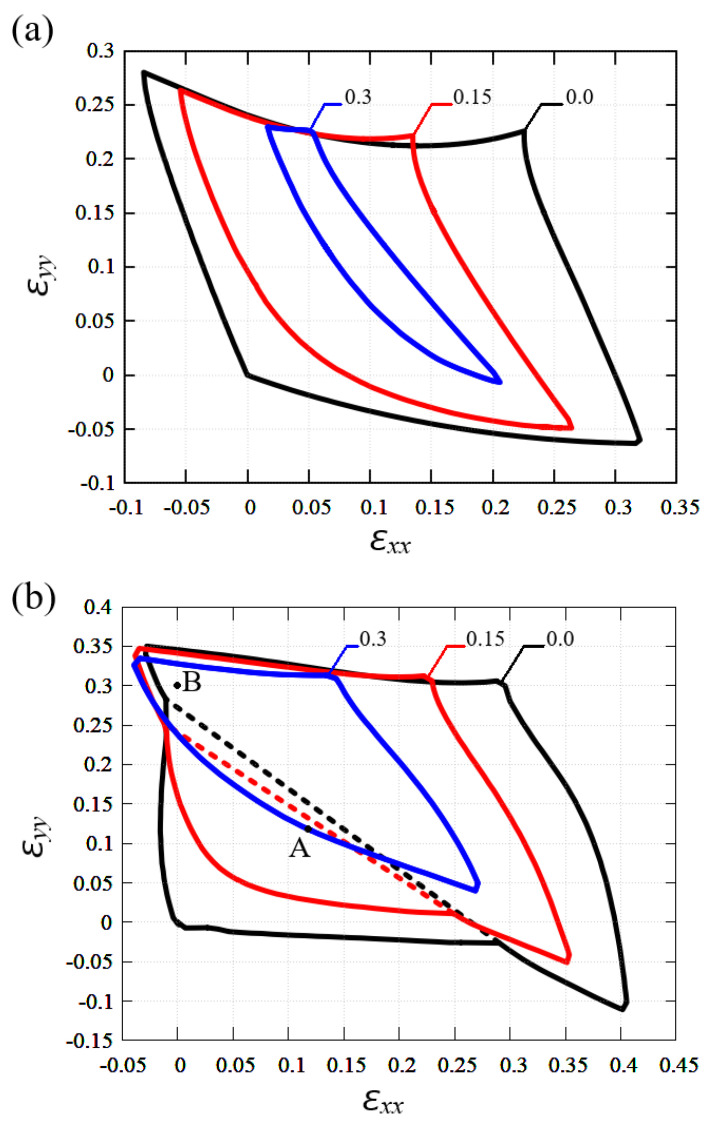
The effect of shear strain on the stability region of planar graphene (**a**); and stanene (**b**). Results for $\varepsilon_{xy}$ = 0.0, 0.15 and 0.3 are plotted by the black, red, and blue lines, respectively. Panel (**b**) also shows the points A and B chosen for the analysis of the stanene structure.

**Figure 8 materials-15-05900-f008:**
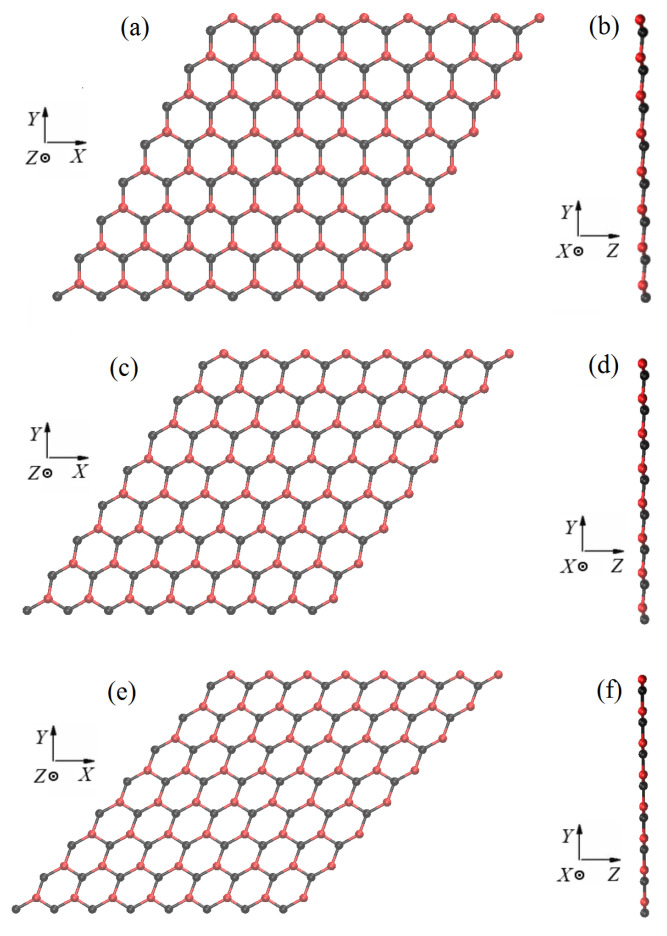
The structure of stanene at point A shown in Figure 7b for the three values of shear strain: (**a**,**b**) for $\varepsilon_{xy}=0$; (**c**,**d**) for $\varepsilon_{xy}=0.15$, and (**e**,**f**) for $\varepsilon_{xy}=0.3$. Other components of strain are $\varepsilon_{xx}=\varepsilon_{yy}=0.118$.

**Figure 9 materials-15-05900-f009:**
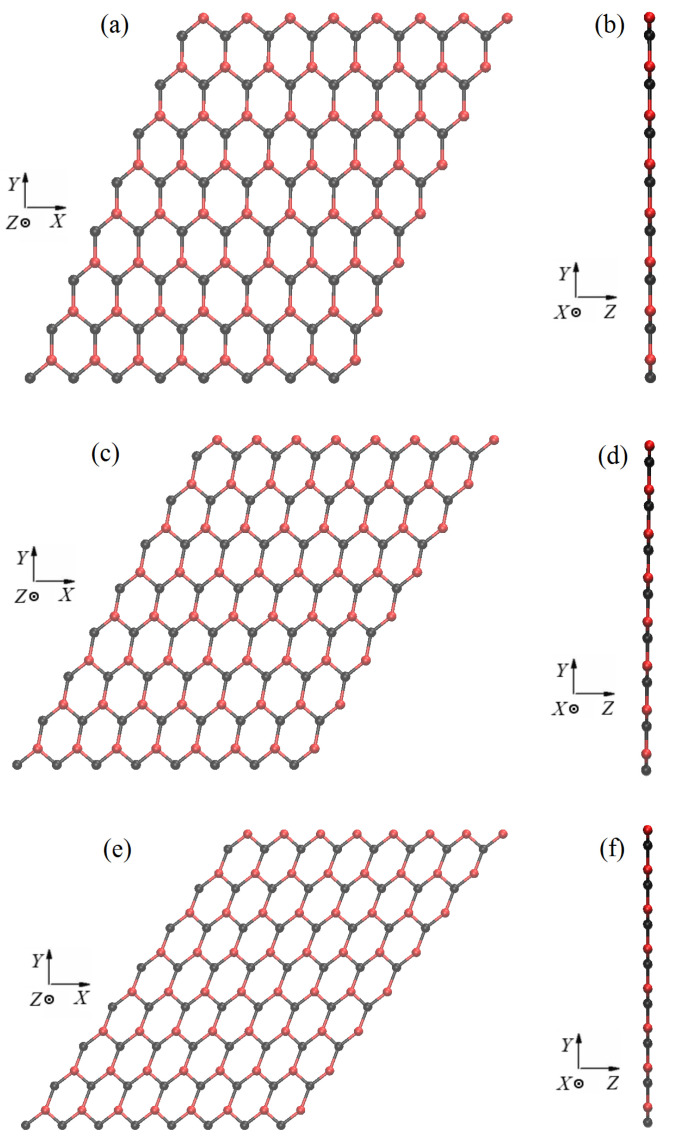
The structure of stanene at point B shown in Figure 7b for the three values of shear strain: (**a**,**b**) for $\varepsilon_{xy}=0$; (**c**,**d**) for $\varepsilon_{xy}=0.15$, and (**e**,**f**) for $\varepsilon_{xy}=0.3$. Other components of strain are $\varepsilon_{xx}=0.0$, $\varepsilon_{yy}=0.3$.

**Table 1 materials-15-05900-t001:** Parameters of the Tersoff potential [66].

*M*	3.0
γ	1.0
$\lambda_{3}$ (Å${}^{-1}$)	0.198543
*c*	501643.0
*d*	155.4496
δ	−0.321475
*N*	1.423982
β	0.006901
$\lambda_{2}$ (Å${}^{-1}$)	0.221467
*B* (eV)	4.534091
*R* (Å)	3.59983
*D* (Å)	0.723836
$\lambda_{1}$ (Å${}^{-1}$)	2.824898
*A* (eV)	638.6396

**Table 2 materials-15-05900-t002:** Parameters of the structures at points A and B marked in Figure 7b. Valence bonds and valence angles are defined in Figure 1b. The potential energy per atom *U* and the height of stanene *h* are also given.

Point	$\varepsilon_{\mathrm{xy}}$	Angles in Degrees			Bond Length in Å			*U* (eV)	*h* (Å)
		$\varphi_{1}=\tau_{3}$	$\varphi_{2}=\tau_{1}$	$\varphi_{3}=\tau_{2}$	1	2	3		
A	0.0	119.20	119.20	119.20	3.027	3.027	3.027	−3.11	0.32
	0.15	119.01	128.44	110.12	3.108	3.038	2.973	−3.06	0.27
	0.3	119.02	138.71	102.27	3.208	3.039	2.911	−2.88	0.0
B	0.0	101.66	130.19	112.15	3.043	3.360	2.989	−2.73	0.0
	0.15	101.37	138.98	119.65	3.090	3.366	2.970	−2.72	0.0
	0.3	100.66	148.08	111.26	3.215	3.381	2.924	−2.60	0.0

## Data Availability

The data and the home-made codes of this study are available from the corresponding author upon reasonable request.

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
