# Peer review of "Stability of Strained Stanene Compared to That of Graphene"

_materials, 2022, doi:10.3390/ma15175900_

Round 1

Reviewer 1 Report

The manuscript titled “Stability of strained stanine and graphene” described molecular dynamics simulation study on mechanical stability of 2D stanene. The manuscript clearly demonstrated comprehensive simulation results and discussion, and comparison with graphene as well, therefore, I would recommend the acceptance in present form in Materials.

Author Response

We are pleased with the positive assessment of our work, thank you.

Reviewer 2 Report

The authors of this study mainly focused on calculating the stability range of stanene and compared their results with previously published reports on graphene. 2D-xenene materials are a growing field of research now and any addition to this research field will be good. However, I cannot recommend this paper in its present form. My comments are as follows:

1.     The authors did not mention clearly about their calculation of graphene. Did they generate the structure of graphene too and carry out calculations on it? If they only compared their results with previously published reports, then the title must be revised.

2.     The authors must describe more about their MD simulation code, as well as they must compare their result with DFT-MD simulation to validate their results.

3.     The authors mentioned they relaxed the structure with “steepest descent method.” What are the criteria? Which code was used for this purpose? They must add comparison of initial and optimized lattice parameters and other structural details after relaxation.  Also, it will be better to add some comparison with previously published reports using DFT or experiments.

4.     The authors must add the phonon dispersion curve for the system. Otherwise, it is difficult to validate the results of interatomic potentials.

5.     Please clarify why do the authors compare Tersoff potential with Savin’s and Brener’s potentials? Did they calculate the Tersoff potential of graphene too? It is not clarified in their ms.

6.     Did the authors calculate Poisson’s ratios? Add the discussion about it.

7.     The authors just presented their calculation results. They must add their views and give reasons to validate their calculated results.

8.     The authors did not declare about the availability of their code. 

Author Response

Dear Reviewers,

We are very grateful for detailed reviews of the manuscript "Stability of strained stanene and graphene" (materials-1852438). We considered all the comments and made appropriate changes to the manuscript. All important changes are highlighted in the manuscript in red. Please see our response and a description of the changes below.

The authors of this study mainly focused on calculating the stability range of stanene and compared their results with previously published reports on graphene. 2D-xenene materials are a growing field of research now and any addition to this research field will be good. However, I cannot recommend this paper in its present form. My comments are as follows:

  1. The authors did not mention clearly about their calculation of graphene. Did they generate the structure of graphene too and carry out calculations on it? If they only compared their results with previously published reports, then the title must be revised.

Our response:

We agree. Since we are comparing our results for stanene with known results for graphene, the new title of the manuscript is "Stability of strained stanene compared to that of graphene".

  1. The authors must describe more about their MD simulation code, as well as they must compare their result with DFT-MD simulation to validate their results.

Our response:

We agree. We have added the following text describing the code used for simulations:

The calculation of dispersion curves and stability analysis for stanene are performed using a home-made C++ code that implements the solution of the eigenvalue problem based on the iterative Jacobi algorithm for calculating the eigenvalues and eigenvectors of a real symmetric matrix. The iterations stop when the maximum off-diagonal element of the stiffness matrix becomes less than 10-5 eV/Å2.

We have added about 20 new references, and many of them describe experimental and DFT results for stanene. The results presented in all of these references are discussed in the Introduction and Results and Discussion sections (highlighted in red).

  1. The authors mentioned they relaxed the structure with “steepest descent method.” What are the criteria? Which code was used for this purpose?

Our response:

The following tex was added:

The relaxation is performed by the steepest descent method using self-made code written in the C++ algorithmic language. Relaxation stops when the value of the maximum force acting on the atoms becomes less than 10-8eV/Å.

  1. They must add comparison of initial and optimized lattice parameters and other structural details after relaxation. Also, it will be better to add some comparison with previously published reports using DFT or experiments.

Our response:

As mentioned in the answer to point 2, a description of the experiments and DFT results for stanene has been added to the Introduction and Conclusions (highlighted in red).

The authors must add the phonon dispersion curve for the system. Otherwise, it is difficult to validate the results of interatomic potentials.

Our response:

We agree. New section 3.2. “Phonon dispersion curves” was added where the dispersion curves are presented and compared to the known DFT results.

  1. Please clarify why do the authors compare Tersoff potential with Savin’s and Brener’s potentials? Did they calculate the Tersoff potential of graphene too? It is not clarified in their ms.

Our response:

In the literature the stability regions of graphene were described for the Savin’s and Brenner’s potentials and we compare our results for stanene with the existing results. The following text was added:

In the literature, the stability regions of graphene were described [48,67] using the Savin potential [49] and the modified Brenner potential [65,68], and here we compare our results for stanene with existing results for graphene. Note that the accuracy of various interatomic potentials for graphene have been assessed in the work [69]. It was proved that the Savin potential reproduces well many properties of carbon structures [70-73].

  1. Did the authors calculate Poisson’s ratios? Add the discussion about it.

Our response:

In the present study, the mechanical properties of stanene were not analyzed.

  1. The authors just presented their calculation results. They must add their views and give reasons to validate their calculated results.

Our response:

The Results and Discussion section was expanded to respond to this critical comment (see text in red).

  1. The authors did not declare about the availability of their code. Eugster, dell'Issola, Steigmann etc.), this investigation suggests a concrete example of a material with unusual properties, and we can see the origin of these properties on the discrete level.

Our response:

We have added the sentence that our codes are available from the authors upon request as follows:

Data Availability Statement: The data and the home-made codes of this study are available from the corresponding author upon reasonable request.

We also appreciate the comment about the work by Eugster et al.; it was cited.

Reviewer 3 Report

The article, „Stability of strained stanene and graphene” focuses on the stability of stanene and its comparison with another 2D material, graphene, by comparing Tersoff, Savin, and Brenner potentials. Stanene is buckled by default and, under stretch, can become flat. Additionally, the existence of two energetically equivalent states was theoretically predicted. The authors are comparing different potential models, which is why the limitations of each of them should be stated. Materials and Methods chapter, please write at least the language of the program you used or the package name, whatever identifies the code used, if commercially available. What is missing is the comparison with experimental methods; the limitation of the taken model should be provided; interactions omitted etc. The article is written thoroughly, and the English language is fine.

Author Response

Dear Reviewers,

We are very grateful for detailed reviews of the manuscript "Stability of strained stanene and graphene" (materials-1852438). We considered all the comments and made appropriate changes to the manuscript. All important changes are highlighted in the manuscript in red. Please see our response and a description of the changes below.

The article, „Stability of strained stanene and graphene” focuses on the stability of stanene and its comparison with another 2D material, graphene, by comparing Tersoff, Savin, and Brenner potentials. Stanene is buckled by default and, under stretch, can become flat. Additionally, the existence of two energetically equivalent states was theoretically predicted.

The authors are comparing different potential models, which is why the limitations of each of them should be stated.

Our response:

We agree. We have added citation of the works where the interatomic potentials were compared:

Note that the accuracy of various interatomic potentials for graphene have been assessed in the work [69]. It was proved that the Savin potential reproduces well many properties of carbon structures [70-73].

Materials and Methods chapter, please write at least the language of the program you used or the package name, whatever identifies the code used, if commercially available.

Our response:

We thank the Reviewer for this critical comment which was also raised by the Reviewer #2. The following information was added:

The calculation of dispersion curves and stability analysis for stanene are performed using a home-made C++ code that implements the solution of the eigenvalue problem based on the iterative Jacobi algorithm for calculating the eigenvalues and eigenvectors of a real symmetric matrix. The iterations stop when the maximum off-diagonal element of the stiffness matrix becomes less than 10-5 eV/Å2.

The relaxation is performed by the steepest descent method using self-made code written in the C++ algorithmic language. Relaxation stops when the value of the maximum force acting on the atoms becomes less than 10-8eV/Å.

Data Availability Statement: The data and the home-made codes of this study are available from the corresponding author upon reasonable request.

What is missing is the comparison with experimental methods; the limitation of the taken model should be provided; interactions omitted etc. The article is written thoroughly, and the English language is fine.

Our response:

The Introduction and the Results and Discussion sections were expanded to respond to this critical comment (see text in red). We have added about 20 new references, and many of them describe experimental and DFT results for stanene. In particular, as the limitations of the Tersoff potential (and also other interatomic potentials) the following was added:

The accuracy of the potential at not very large atomic displacements can be estimated by comparing the phonon dispersion curves shown in Figure 3 with those calculated on the basis of the DFT theory [32]. As already mentioned, the agreement between the dispersion curves calculated using the Tersoff potential and those calculated ab initio is quite good. The accuracy of the Tersoff potential at large values of strain may not be as good because the dispersion curves are calculated for the linearized equations of motion.

Round 2

Reviewer 2 Report

The authors of this study considered most of my comments and modified their ms accordingly. However, I still have some more concerns and I cannot accept this ms in its present form. My comments are as follows:

1.     The authors must provide the equation for energy of their MD-simulation code in detail. Also, they should discuss about the used potentials etc.

2.     I cannot accept their answer to comment 6. They are using shear strain to define the stability region without analyzing the mechanical properties. Why? It is difficult to validate their results without any mechanical properties. I urge the authors to add this and compare their own results with DFT-simulations.

Author Response

Response to the critical comments of Reviewer #2:

 Dear Reviewer #2,

We are very grateful for the second review of the manuscript "Stability of strained stanene and graphene" (materials-1852438). We have addressed both critical comments. All important changes are highlighted in the manuscript in blue.

  1. The authors must provide the equation for energy of their MD-simulation code in detail. Also, they should discuss about the used potentials etc.

Our response:

We have expanded Sec. 2.3 on pages 5 and 6 describing the Tersoff potential and its parameters. The Hamilton Equation (8) was added. We believe that the information provided is sufficient to reproduce our results.

  1. I cannot accept their answer to comment 6. They are using shear strain to define the stability region without analyzing the mechanical properties. Why? It is difficult to validate their results without any mechanical properties. I urge the authors to add this and compare their own results with DFT-simulations.

Our response:

As per request of the Reviewer #2 we have calculated the Poisson’s ratio and compared it with DFT results (see text in blue on pages 7 and 8). It is tempting to add more results to this manuscript, but we would like to focus on our main result, which is to demonstrate the possibility of the existence of domain walls separating the +Z and –Z domains.

***************************************************************************

Kind regards,

I.V. Kosarev, S.V. Dmitriev, A.S. Semenov, E.A. Korznikova

Round 3

Reviewer 2 Report

The authors of this paper modified their paper based on my comments. I can not see any other improvement from my side, hence suggest publication of this work. 

On a minor note, many numbers while citing references or figures became "?" in the last version of ms. I think it is due to some technical error. Please make a careful check before publication.